# Electrochromic Properties of Li- Doped NiO Films Prepared by RF Magnetron Sputtering

**Jui-Yang Chang [1], Ying-Chung Chen [1], Chih-Ming Wang [2,*] and You-Wei Chen [1]**

[1] Department of Electrical Engineering, National Sun Yat-sen University, Kaohsiung 804, Taiwan; twm0929137151@gmail.com (J.-Y.C.); ycc@mail.ee.nsysu.edu.tw (Y.-C.C.); s8689101522@gmail.com (Y.-W.C.)

[2] Department of Electrical Engineering, Cheng Shiu University, Kaohsiung 833, Taiwan

\* Correspondence: cmwang@gcloud.csu.edu.tw

**Abstract:** In this study: various amounts of $Li_2CO_3$ powders were mixed into NiO powders to fabricate the Li- added NiO (NiO:Li) targets. The electrochromic films of LiNiO were deposited on ITO glasses at room temperature (R.T.) by RF magnetron sputtering. The thicknesses of electrochromic LiNiO films were kept about 200 nm. The ECD device was constructed with structure of Glass/ITO/ LiNiO /Gel-electrolyte/ITO/Glass. The results indicated that the optimal electrochromic characteristics of $Li_{0.16}Ni_{0.58}O$ thin films could be obtained by 10 wt% $Li_2CO_3$ added NiO target. The optimized characteristics of ECDs could be achieved with the intercalation charge (Q) of 11.93 mC/cm$^2$, the optical density ($\Delta$OD) of 0.38, the transmittance change ($\Delta$T) of 44.1%, and the coloring efficiency ($\eta$) of 31.8 cm$^2$/C at the wavelength of 550 nm by setting voltage of 3.2V. The results demonstrate that the doping of Li$^+$ ions into NiO films can effectively enhance the characteristics of ECD devices. The reason may due to the increased amount of charge stored in the electrochromic devices (ECDs).

**Keywords:** Nickel oxide: Lithium carbonate; RF magnetron sputtering; Electrochromic device

---

## 1. Introduction

Facing the growing energy crisis, studies have suggested that establishing the energy-saving system is more effectively than developing new energy sources continuously. Electrochromic glasses have been investigated widely and expected to substitute the traditional energy-saving glasses [1,2]. Electrochromic devices can be applied in many fields, such as, smart windows, sunroofs of automobiles, car rear-view mirrors, sunglasses, and etc. [3,4]. In 1961, Platt first reported the electrochromic phenomenon, in which, a new light absorption band would be resulted in a material due to the transfer of electrons through redox or migration reactions, which will lead to the color changes of materials [5]. Among the electrochromic materials, nickel oxide films have been intensively researched as electrochromic films because of their large optical density and satisfactory cycling reversibility.

In general, the ECD device is constructed as Glass/ITO/NiO/electrolyte/ITO/Glass, Glass/ITO/WO$_3$/electrolyte/ITO/Glass or complementary electrochromic device as (Glass/ITO/NiO/electrolyte/WO$_3$/ITO/Glass, CECD). The ceramic oxides used as electrochromic materials can be classified into the anodically coloring materials, such as NiO, Rh$_2$O$_3$, CoO$_2$ and IrO$_2$, and the cathodically coloring materials, such as WO$_3$, TiO$_2$, Nb$_2$O$_5$ and MoO$_3$. The color changes of materials are the results of intercalation/deintercalation for both electrons (or hole) and cations (or anions). By comparing, WO$_3$ film exhibited better efficiency and ion storage capacity than those of NiO film. If the coloring characteristics and ion storage capacity of the NiO thin film can be enhanced, the application of the ECD or CECD devices must be greatly promoted. According to the literatures, the characteristics of electrochromic films can be effectively enhanced by Li-doping [6–11]. The Li$^+$ ions can provide fast charge transfer carriers as ion transportation in the ECDs [12].

In this study, various amounts of lithium carbonate (Li2CO3) powders were mixed into the nickel oxide (NiO) powders to obtain the NiO:Li targets. Electrochromic thin films were deposited on indium tin oxide (ITO) glasses with the optimized fabrication parameters by RF magnetron sputtering. The gel polymer electrolyte (GPE) containing lithium perchlorate was synthesized to be an ion storage layer. As a result, the electrolyte and electrochromic films in the ECDs both contain lithium ions which will elevate the rate of charge transfer and enhance the characteristics of ECDs. The amount of charge storage was expected to be raised by doping of $Li^+$ into NiO films, which makes ECDs operating optimally. In accordance, the ECD devices with the structure of Glass/ITO/LiNiO/electrolyte/ITO/Glass were fabricated, and the characteristics were investigated, such as, cyclic voltammetry (CV), transmittance spectra, optical density, coloring efficiency and etc. to obtain the optimized ECD devices.

## 2. Experimental

### 2.1. Preparation of NiO:Li Targets

Various amounts of lithium carbonate (SIGMA-ALDRICH, 98%, $Li_2CO_3$) powders from 0 wt.% to 15 wt.% were mixed into NiO (Alfa Aesar, 99%) powders in separate experiments, then the mixed powders were calcined at 900°C for 24 h to obtain the LiNiO powders. After grinding, the calcined LiNiO powders were filled into a die and pressed to obtain a pellet with diameter of 2 inches and thickness of 1 cm. The pellet was sintered at 1100°C for 24 h to obtain the desired targets. The targets with various weight of $Li_2CO_3$ powders from 0 wt.% to 15 wt.%, were named as NiO, LiNiO 5, LiNiO 10, LiNiO 15, as shown in Table 1. The photographs of prepared 2 inches targets are shown in Figure 1.

**Table 1.** The names of targets with various $Li_2CO_3$ amounts.

| $Li_2CO_3$ Amount. (wt%) | Name |
|---|---|
| 0 | NiO |
| 5 | LiNiO 5 |
| 10 | LiNiO 10 |
| 15 | LiNiO 15 |

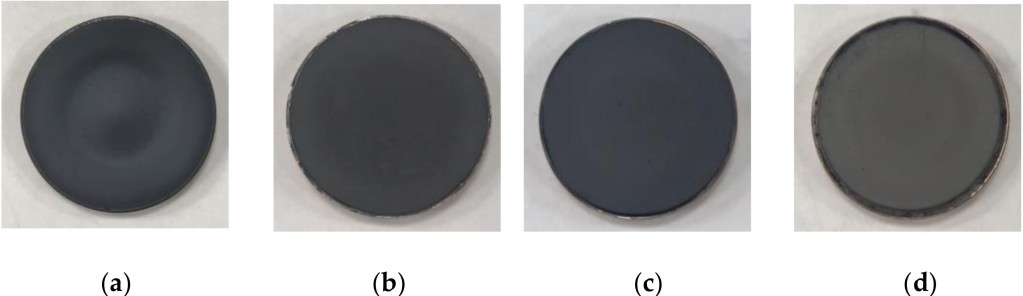

| (a) | (b) | (c) | (d) |

**Figure 1.** The prepared targets: (**a**) NiO, (**b**) LiNiO 5, (**c**) LiNiO 10 and (**d**) LiNiO 15.

### 2.2. Preparation of ECD s

In this study, the electrochromic LiNiO films were deposited on ITO glasses with size of 3 cm × 3.5 cm by RF magnetron sputtering. The sheet resistance of ITO film was about 7 Ω, and the thickness was about 250 nm. Figure 2 shows the transmittance spectra of ITO. The substrates were placed underneath the target. The base pressure of chamber was about $6.5 \times 10^{-4}$ Pa. The distance of substrate from target was 5 cm. The deposition parameters of LiNiO films are shown in Table 2.

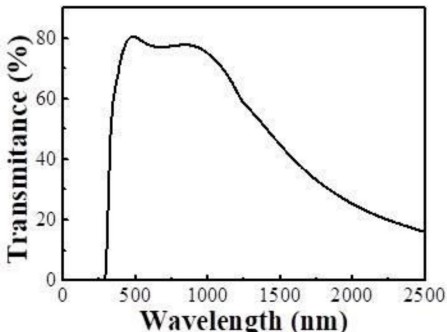

**Figure 2.** The transmittance spectra of ITO glass.

**Table 2.** Deposition parameters of LiNiO films.

| Target | NiO:Li |
|---|---|
| Substrate | ITO/Glass |
| Substrate rotation speed (rpm) | 40 |
| Base pressure (Pa) | $<6.5 \times 10^{-4}$ |
| Sputtering power (W) | 100 |
| Deposition pressure (Pa) | 1.7 |
| Deposition temperature (°C) | R.T. |
| Oxygen concentration ($O_2$/Ar+$O_2$, %) | 65 |
| Film thickness (nm) | 200 |

The ECD device was constructed as Glass/ITO/LiNiO/GPE/ITO/Glass. The device was fabricated combined GPE and electrochromic films by hot pressing processes, with the pressure of 0.5 MPa for 30 s, followed by the pressure of 1.0 MPa for 30 s and finally by the pressure of 1.5 MPa for 30 s. The photograph of a fabricated ECD sample is shown in Figure 3.

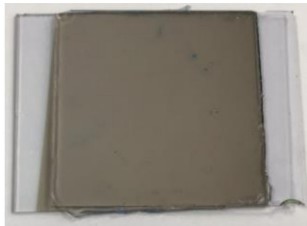

**Figure 3.** Photograph of a fabricated ECD sample.

*2.3. Characteristics Analysis*

X-ray diffraction (XRD, Bruker D8 Advance, Billerica, MA, USA) with Cu-K$\alpha$ radiation ($\lambda$ = 0.1542 nm) was adopted to analyze the crystalline structures of the LiNiO films. The scanning rate of 4°/min was performed in the 2$\theta$ range from 20° to 60°. The scanning electron microscope (SEM, ZEISS, Auriga-39-50, Oberkochen, Germany) was used to investigate the surface morphologies of the electrochromic films. The measurements of electrochromic properties were carried out in a two-electrode cell with an electrochemical analyzer (CHI, 6273B, Austin, USA), in which, the ITO/Glass acted as the reference electrode and the counter electrode simultaneously, and the LiNiO/ITO/Glass acted as the working electrode. The measurements of cyclic voltammetry (CV) were carried out ranging from +3.2 to −3.2 V with the sweep rate of 50 mV/s. An ultraviolet visible near-infrared spectrophotometer (JASCO, V-570) was used to measure the optical transmittance spectra of ECDs in the range from 200 to 2500 nm wavelength. Electron spectroscopy for Chemical Analysis (ESCA) is adopted to analyze the elementary composition of the film surfaces [13].

## 3. Results and discussion

Table 3 and Figure 4 show the stoichiometric ratio of NiO and LiNiO films. It is obvious that when the added amount of $Li_2CO_3$ in the target increased, the atomic percentages of Li and O increased, whereas the Ni atomic percentage decreased. As can be seen in Figure 4, with the increase of added amount of $Li_2CO_3$ in the target, Ni/O ratio decreased from 1.03 to 0.37 and (Li+Ni)/O ratio decreased from 1.03 to 0.59, the results revealed that Ni indeed has been substituted by Li. The LiNiO films deposited by different targets were named as Films A, B, C and D.

**Table 3.** Chemical analysis of LiNiO films.

| Target | Li (at.%) | Ni (at.%) | O (at.%) | LiNiO Films |
|--------|-----------|-----------|----------|-------------|
| NiO | - | 50.73 | 49.27 | $Ni^{1.03}O$ (denoted as Film A) |
| LiNiO 5 | 6.59 | 39.53 | 53.88 | $Li^{0.12}Ni^{0.73}O$ (denoted as Film B) |
| LiNiO 10 | 9.39 | 33.11 | 57.5 | $Li^{0.16}Ni^{0.58}O$ (denoted as Film C) |
| LiNiO 15 | 13.85 | 23.27 | 62.88 | $Li^{0.22}Ni^{0.37}O$ (denoted as Film D) |

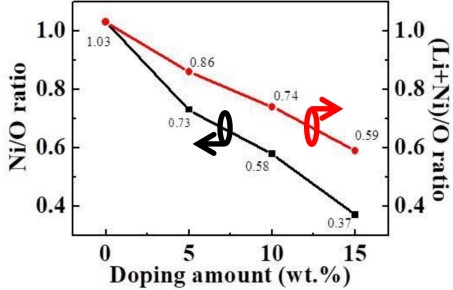

**Figure 4.** Stoichiometric ratio of LiNiO films.

Figure 5 shows the XRD patterns of Films A, B, C and D deposited on the glass substrates. The results show that all the films are amorphous, which might be due to the insufficient energy to induce crystalline growth of electrochromic films deposited at R.T. [14]. The amorphous state is desired because it is easier for the ion transportation in the ECDs.

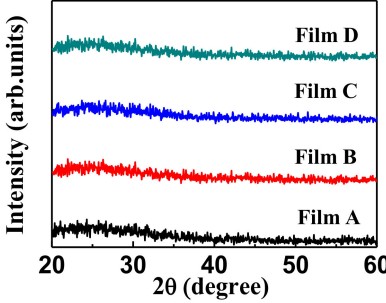

**Figure 5.** The XRD patterns of electrochromic films A, B, C and D.

Figure 6 shows the top views of electrochromic films deposited on the ITO substrates. Figure 4a,b show that the surfaces of electrochromic films existed some slightly irregular boundaries. In Figure 6c, some massive grains, which will be identified and discussed below, on the surface of the Film C can be found [8]. In Figure 6d, there are no obvious massive grains on the surface, the possible reason might be due that the sputtering rate of LiNiO 15 target is too slow.

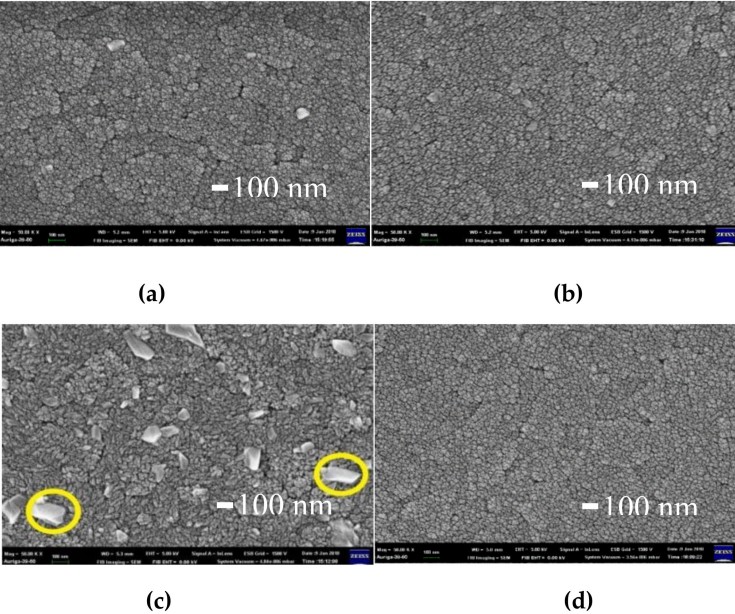

**Figure 6.** Surface morphologies of electrochromic films: (**a**) Film A, (**b**) Film B, (**c**) Film C and (**d**) Film D.

Figure 7 shows the XPS analysis of O1s on the surfaces of electrochromic Films A and C. As shown in Figure 5a, the O1s binding energy on the surface of NiO film was 529.5 eV, whereas in Figure 7b, the O1s binding energy appeared in two peaks, which were 529.5 eV and 531.5eV, and the latter was more evident. According to the literature [4], the O1s binding energies of NiO and $Ni_2O_3$ were 529.5 eV and 531.5 eV, which inferred that the massive grains on the surface of electrochromic film shown in Figure 6c might be the formed crystalline particles of $Ni_2O_3$.

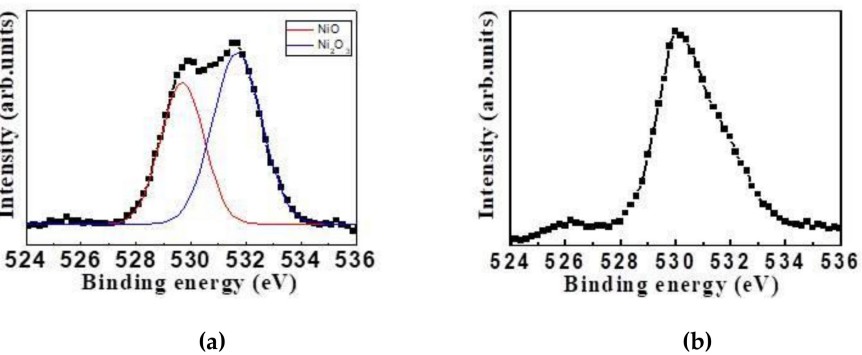

**Figure 7.** XPS analysis of O1s on the film surfaces: (**a**) Film A, and (**b**) Film C.

Figure 8 demonstrates the cyclic voltammetric (CV) characteristics of ECDs with a bleaching voltage of -3.2 V and a coloring voltage of 3.2 V. The scanning loop starts from 0 V→ 3.2 V→ 0 V→ -3.2 V → 0 V in sequence with the scanning rate of 50 mV/s. The peak at 0 V indicates that $Li^+$ ions are difficult to be absorbed by the counter electrode of ITO glass, and will migrate to the working electrode or electrochromic films [11]. At the peak of 3.2 V, the ions and the electrons will intercalate into electrochromic films. In contrast, at the peak of -3.2 V, the ions and electrons will be expelled. The optimal operating voltages of the devices are -3.2 V and 3.2 V The area of CV of NiO film is smaller than those of LiNiO films, which means that the electrical conductivity could be increased by doping of $Li^+$ [15]. The results indicate that doping of $Li^+$ into the electrochromic films will improve the intercalation/deintercalation reactions of $Li^+/e^-$ during the cyclic voltammetric processes.

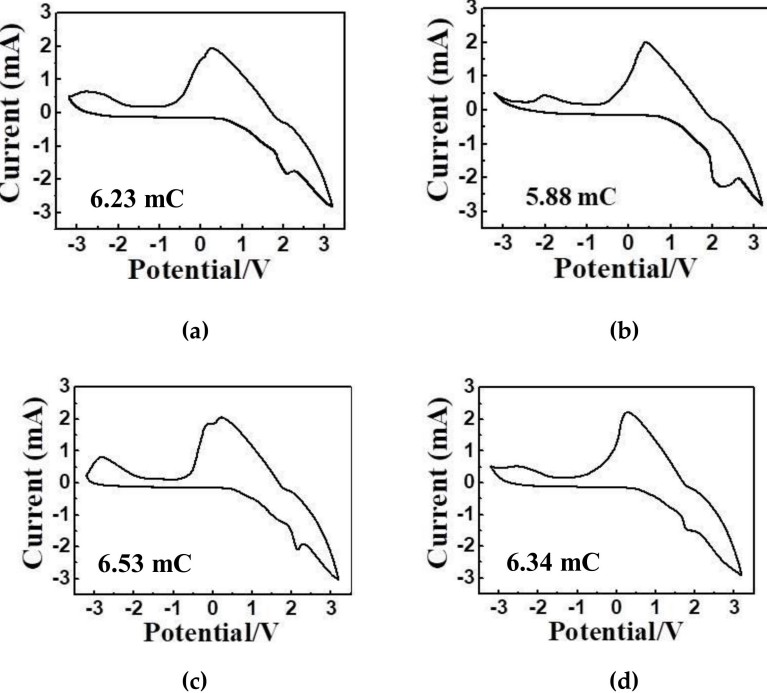

**Figure 8.** Cyclic voltammetic characteristics of ECDs: (**a**) Film A, (**b**) Film B, (**c**) Film C and (**d**) Film D.

Figure 9 shows the transmittance spectra of ECDs with various doping amounts of $Li^+$. The spectra were obtained after the coloring and bleaching voltages of 3.2 V and $-$3.2 V were applied to the ECDs and last for 12 seconds, respectively. Table 4 shows the transmittance variations ($\Delta T$) and optical density variations ($\Delta OD$) of ECDs with various electrochromic films. $\Delta T$ is defined as $T_{bleached} - T_{colored}$, and $\Delta OD$ is given as $\Delta OD = \log (T_{bleached}/T_{colored})$ at 550 nm, respectively. The results indicate that the optical properties of ECDs will be varied with the doping amount of $Li^+$. With the increase of $Li^+$, both $\Delta T$ and $\Delta OD$ were significantly increased. The reason might be due to the increased conductivity of films with the increased doping amount of $Li^+$ [15]. It might also improve the intercalation/deintercalation reactions of $Li^+/e^-$ during the coloring/bleaching processes. The transmittance spectra shows that ECD with NiO film without doping exhibits the lowest $\Delta T$ and $\Delta OD$, whereas the optimized $\Delta T$ of 44.1% and $\Delta OD$ of 0.38 at 550 nm can be obtained in ECD with Film C. However, when an excessive amount of $Li^+$ doped into NiO, the $\Delta T$ and $\Delta OD$ decreased. The over doping of $Li^+$ ions will lead to the lower carrier mobility and electrical conductivity of electrochromic films [16]. According to Stafstrom et al., it is appropriate for monitors and smart glasses when $\Delta OD$ is above 0.3 [17,18].

The coloration efficiency is defined as the variation of optical density per unit of charges inserted into or extracted from the electrochromic material in a given wavelength. It is one of the most important parameters in the evaluation of electrochromic materials. In Table 4, the coloration efficiency increases with the increased doping amount of $Li^+$, however, a decrease in the coloration efficiency would be caused by the excessive doping of $Li^+$. The Film C exhibits the optimized coloration efficiency of 31.8 $cm^2/C$. It was pointed out that the electrical conductivity of electrochromic films can be increased by doping of Li+, whereas, excessive doping of $Li^+$ ions will result in a lower carrier mobility and decreased electrical conductivity and also the decreased coloration efficiency of ECDs [13]. Details of the characteristics of the electrochromic thin films are shown in Table 4.

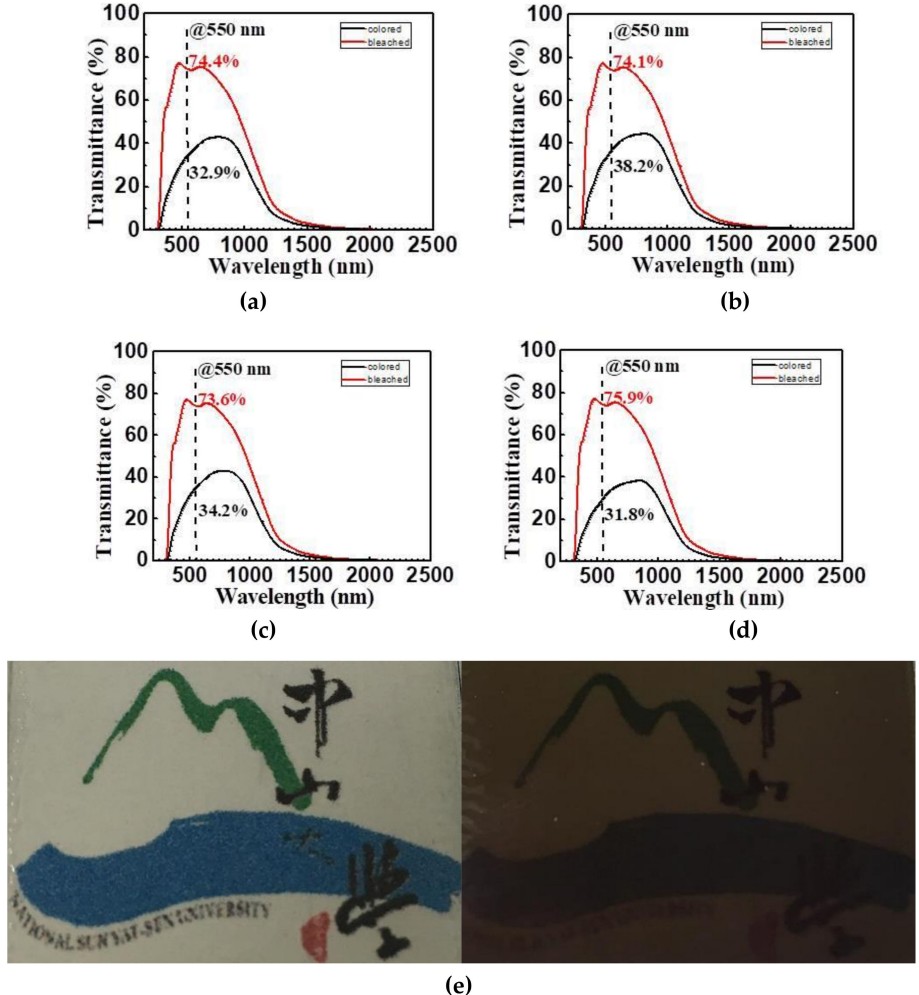

**Figure 9.** Transmittance spectra of ECDs: (**a**) Film A, (**b**) Film B, (**c**) Film C and (**d**) Film D, and (**e**) the photographs of the Film C in coloring and bleaching states.

**Table 4.** The electrochromic properties of ECDs ($\lambda$ = 550 nm).

| Film | Bleaching Transmittance ($T_b$, %) | Coloring Transmittance ($T_c$, %) | Transmittance Change ($\Delta T$, %) | Intercalation Charge ($Q$, mC/cm$^2$) | Optical Density ($\Delta OD$) | Coloration Efficiency ($\eta$, cm$^2$/C) |
|---|---|---|---|---|---|---|
| Film A | 74.1 | 38.2 | 35.9 | 11.39 | 0.29 | 25.4 |
| Film B | 74.4 | 32.9 | 41.5 | 11.70 | 0.35 | 29.9 |
| Film C | 75.9 | 31.8 | 44.1 | 11.93 | 0.38 | 31.8 |
| Film D | 73.6 | 34.2 | 39.4 | 12.45 | 0.33 | 26.5 |

## 4. Conclusions

In this study, the electrochromic LiNiO films were deposited on ITO glasses at R.T. by RF magnetron sputtering technology. The cyclic voltammetry analysis of electrochromic films showed that the hysteresis area of CV increased as Li$^+$ ions were doped into NiO. However, an excessive doping amount of Li$^+$ will result in the worse characteristics of ECD devices. The ECD device with Li$_{0.16}$Ni$_{0.58}$O film as the electrochromic material exhibits the optimized characteristics, in which, the $\Delta T$ of 44.1%, the $\Delta OD$ of 0.38, the Q value of 11.93 mC/cm$^2$, and the $\eta$ of 31.8 cm$^2$/C were obtained at 550 nm wavelength. The results demonstrate that Li$^+$ doping can effectively improve the electrochromic properties of NiO films and will be suitable for the applications of smart windows, and etc.

**Author Contributions:** Data curation, J.-Y.C., C.-M.W. and Y.-W.C.; Formal analysis, J.-Y.C.; Writing – original draft, J.-Y.C. and Y.-W.C.; Writing – review & editing, Y.-C.C. and C.-M.W. All authors have read and agreed to the published version of the manuscript.

**Funding:** This study was supported by the Ministry of Science and Technology of the Republic of China, Taiwan (No. MOST 107-2221-E-230-006).

**Conflicts of Interest:** The authors declare no conflict of interest.

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
