# Peer review of "Electrochromic Properties of Li- Doped NiO Films Prepared by RF Magnetron Sputtering"

_coatings, doi:10.3390/coatings10010087_

Round 1

Reviewer 1 Report

Minor remarks:

1. Introduction
- please provide  state of art for the electrochromic achievments with recently published references

2. Experimental
- please add the photos of the prepared targets in section 2.1
- table 2, what does it mean (O2/Ar+O2, %) ?
- please add a photo of the deposited samples ECD

3. Results and Disscusion
- please add the table where differences between samples A,B,C,D will be summarized (or add columns in table 4)

Author Response

Minor remarks:

Introduction
- please provide  state of art for the electrochromic achievments with recently published references

Ans: Thanks for reviewer’s comment. The more references are added in line 27~29. ECDs have many applications including: architectural smart windows, car rear-view mirrors, sunroofs of automobiles, view angle-independent display, sunglasses, etc...[3,4].

Experimental
- please add the photos of the prepared targets in section 2.1
- table 2, what does it mean (O2/Ar+O2, %) ?
- please add a photo of the deposited samples ECD

Ans: Thanks for reviewer’s comment. The photos of the prepared targets added in line 66~71. The (O2/Ar+O2, %) means when deposited the films by rf magnetron sputtering we use the various ratio of Ar and O2 gas into the chamber in the same time. The photos of the deposited samples ECD added in line 94~97.

Results and Disscusion
- please add the table where differences between samples A,B,C,D will be summarized (or add columns in table 4)

Ans: Thanks for reviewer’s comment. The more descriptions are added in line 118~119. And the LiNiO films were deposited by different targets and named Films A, B, C and D.

Reviewer 2 Report

This manuscript describes the introduction of Li-ion in NiO powders to enhance the electrochromic properties. This manuscript may be accepted after minor revision.

1/ There are several abbreviations introduced in abstract which is not use in abstract itself. Please introduce only abbreviations in abstract if the author will use them. The abstract is not attractive as it only shows the obtained results.

2/ Line 25: Can the authors highlight how electrochromic materials can replace the traditional energy-saving glasses?

3/ Line 31-33: The authors are advised to use ECD and CECD abbreviation instead of long stacked mutlilayers in manuscript. The author should provide a summary in table with several materials/ECD, …

4/ What is different between Li-doping and Li-added? Please introduce one term.

5/ In this manuscript, the author have introduced Li-ions to improve electrochromic properties. However, is there also another options than Li-ions? Why was Li-ions chosen prior to another ions?

6/ Experimental section: can the authors provide more information of NiO and LiCO3 powders (Supplier, purify etc). The mixed powder was calcined at 900 °C under which conditions? Same for the sintering of pellets at 1100°C.

7/ Line 70: What is the mean of this abbreviation GPE?

8/ The results of ESCA are shown on Table 3. What is the measurement error in this technique?

9/ Line 130. The authors have described SEM images in the text but it is not clear to the reader what the authors mean with “slightly irregular slits and non-significant crystal particles” and “massive grains”. Can the authors mark them on SEM images (Figure 4)?

10/ Figure 6 is still unclear to the readers as it is difficult to distinguish these scanning loops. The authors are advised to reconsider this part.

11/ It is difficult to follow the indications of the samples (Film B – LiNiO5 – Li0.12Ni0.73O). The authors should introduced one term for one sample.

12/ After a careful read of discussion, it remains unclear what the authors want to do in the next step as it is clear shown that LiNiO 10 modules exhibit good yield. What is the suggestion for the next steps?

Author Response

1/ There are several abbreviations introduced in abstract which is not use in abstract itself. Please introduce only abbreviations in abstract if the author will use them. The abstract is not attractive as it only shows the obtained results.

Ans: Thanks for reviewer’s comment. The more descriptions are added in line 14~15 and corrected several abbreviations. It was expected that the amount of charge stored in the electrochromic devices (ECDs) could be enhanced by using the doping method in the cathode materials.

2/ Line 25: Can the authors highlight how electrochromic materials can replace the traditional energy-saving glasses?

Ans: Thanks for reviewer’s comment. The more descriptions are added in line 27~29. ECDs have many applications including: architectural smart windows, car rear-view mirrors, sunroofs of automobiles, view angle-independent display, sunglasses, etc...[3,4]. Details of advantage and disadvantage showed in table.

ECD

Traditional energy-saving glasses (Low-E glasses)

Advantage

1. Visible light transmittance: ~75%

2. Control the transmittance by yourself

3. Block infrared ~95%

4. Low driving voltage

1. Visible light transmittance: ~75%

2. Block infrared ~75%

Disadvantage

1.Fabricated complexly

1.Fabricated complexly

2.No control the transmittance

3/ Line 31-33: The authors are advised to use ECD and CECD abbreviation instead of long stacked mutlilayers in manuscript. The author should provide a summary in table with several materials/ECD, …

Ans: Thanks for reviewer’s comment. The more descriptions are added in line 36~40 and corrected several abbreviations. Inorganic ceramics oxides in electrochromic materials can be cathodically colored materials: MoO3, WO3, TiO2 and Nb2O5 belong to n-type semiconductors, the anodically colored materials: NiO, IrO2, Rh2O3 and CoO2 belong to p-type semiconductors, are a result of the injection or extraction of both a cation (or anion) and an electron (or hole).

4/ What is different between Li-doping and Li-added? Please introduce one term.

Ans: Thanks for reviewer’s comment. In this study, we use Li-added means the targets were added more Li2CO3 powder into the NiO powder, but when deposited on ITO glasses and measured the chemical analysis of LiNiO films that we got Li was not too much.

5/ In this manuscript, the author have introduced Li-ions to improve electrochromic properties. However, is there also another options than Li-ions? Why was Li-ions chosen prior to another ions?

Ans: Thanks for reviewer’s comment. According to the reference, the Li+ ions can provide fast charge transfer carriers as ion transportation in the electrochromic devices. In this study, the gel polymer electrolyte containing lithium perchlorate was synthesized to be an ion storage layer. In this structure of electrochromic device, both electrolyte and electrochromic films contain lithium ions that will enhance the rate of charge transfer to improve the ECD characteristics.)

6/ Experimental section: can the authors provide more information of NiO and LiCO3 powders (Supplier, purify etc). The mixed powder was calcined at 900 °C under which conditions? Same for the sintering of pellets at 1100°C.

Ans: Thanks for reviewer’s comment. The more descriptions of NiO and Li2CO3 powders (Supplier, purify) are added in line 57~58. Both of the conditions of 900°C and 1100°C were same in heating rate of 1.5°C/min and holding temperature for 24 hours.

7/ Line 70: What is the mean of this abbreviation GPE?

Ans: Thanks for reviewer’s comment. The more descriptions are added in line 48. The GPE means the gel polymer electrolyte.

8/ The results of ESCA are shown on Table 3. What is the measurement error in this technique?

Ans: Thanks for reviewer’s comment. The measurement error in this technique is absolute error of ±2 atomic%

9/ Line 130. The authors have described SEM images in the text but it is not clear to the reader what the authors mean with “slightly irregular slits and non-significant crystal particles” and “massive grains”. Can the authors mark them on SEM images (Figure 4)?

Ans: Thanks for reviewer’s comment. The more descriptions are corrected in line 149~150. Figures 4(a) and (b) show that the surfaces of electrochromic films existed some slightly irregular boundary, and we marked the massive grains on SEM images.

10/ Figure 6 is still unclear to the readers as it is difficult to distinguish these scanning loops. The authors are advised to reconsider this part.

Ans: Thanks for reviewer’s comment. The more descriptions are added in line189~190. The scanning loop starts from 0 V, 3.2 V, 0 V, -3.2 V to 0 V in sequence with the scanning rate of 50 mV/s.

11/ It is difficult to follow the indications of the samples (Film B – LiNiO5 – Li0.12Ni0.73O). The authors should introduced one term for one sample.

Ans: Thanks for reviewer’s comment. The more descriptions are added in line 118~119. And the LiNiO films were deposited by different targets and named Films A, B, C and D.

12/ After a careful read of discussion, it remains unclear what the authors want to do in the next step as it is clear shown that LiNiO 10 modules exhibit good yield. What is the suggestion for the next steps?

Ans: Thanks for reviewer’s comment. We will research the optimized characteristics the electrochromic LiWO3 films by rf magnetron sputtering and finally to combine complementary electrochromic device (Glass/ITO/LiNiO/electrolyte/LiWO3/ITO/Glass).
